# The Role of Inflammation and Myeloperoxidase-Related Oxidative Stress in the Pathogenesis of Genetically Triggered Thoracic Aortic Aneurysms

**DOI:** 10.3390/ijms21207678

**Published:** 2020-10-16

**Authors:** Cassandra Malecki, Brett D. Hambly, Richmond W. Jeremy, Elizabeth N. Robertson

**Affiliations:** 1Discipline of Pathology and Charles Perkins Centre, The University of Sydney, Sydney, NSW 2006, Australia; brett.hambly@sydney.edu.au (B.D.H.); richmond.jeremy@sydney.edu.au (R.W.J.); elizabeth.robertson@sydney.edu.au (E.N.R.); 2Cardiology Department, Royal Prince Alfred Hospital, Sydney, NSW 2050, Australia

**Keywords:** thoracic aortic aneurysm, myeloperoxidase, inflammation, oxidative stress, matrix metalloproteinases

## Abstract

Genetically triggered thoracic aortic aneurysms (TAAs) are usually considered to exhibit minimal levels of inflammation. However, emerging data demonstrate that specific features of an inflammatory response can be observed in TAA, and that the extent of the inflammatory response can be correlated with the severity, in both mouse models and in human studies. Myeloperoxidase (MPO) is a key mediator of the inflammatory response, via production of specific oxidative species, e.g., the hypohalous acids. Specific tissue modifications, mediated by hypohalous acids, have been documented in multiple cardiovascular pathologies, including atherosclerosis associated with coronary artery disease, abdominal aortic, and cerebral aneurysms. Similarly, data are now emerging that show the capacity of MPO-derived oxidative species to regulate mechanisms important in TAA pathogenesis, including alterations in extracellular matrix homeostasis, activation of matrix metalloproteinases, induction of endothelial dysfunction and vascular smooth muscle cell phenotypic switching, and activation of ERK1/2 signaling. The weight of evidence supports a role for inflammation in exacerbating the severity of TAA progression, expanding our understanding of the pathogenesis of TAA, identifying potential biomarkers for early detection of TAA, monitoring severity and progression, and for defining potential novel therapeutic targets.

## 1. Introduction

Aneurysms, defined as the permanent dilation of an artery to a diameter 1.5 times greater than normal size, predispose to the potentially fatal events of dissection or rupture. Based on anatomical location, there are broadly two types of aortic aneurysms, abdominal aortic aneurysms (AAA) and thoracic aortic aneurysms (TAA). Both types of aneurysms reflect underlying structural impairment of the aortic wall. Whilst clear genetic correlates have been difficult to establish in AAA, the pathogenesis of which appears to be environmentally influenced, there is a strong genetic influence in the pathogenesis of some forms of TAA [1].

The causes of TAA include age-related degenerative changes within the aortic wall, thought to be largely secondary to environmental risk factors, such as hypertension, smoking, and atherosclerosis [2]. A substantial number of TAA occur in younger patients, and a genetic pathogenesis has either been proven, or is highly likely, for these patients [3], e.g., Marfan syndrome (MFS), Loeys–Dietz syndrome (LDS), and vascular Ehlers–Danlos syndrome (vEDS). TAA may also be caused by infective, granulomatous, and autoimmune disorders, which are beyond the scope of this review to consider [4].

In younger patients with TAA, familial syndromal and non-syndromal forms have been identified, with a causative gene mutation identified in 20% of TAA [3]. Mutations show wide heterogeneity, with over 40 genes associated with TAA [5]. Identified genes affect diverse protein systems, including extracellular matrix (ECM) regulation (*FBN1, FBN2, COL3A1*), vascular smooth muscle cell (VSMC) contractile apparatus (*MYH11, ACTA2*), and transforming growth factor-beta (TGF-β) signaling (*TGFB2, TGFBR1, SMAD3*). Each of these protein systems has an essential role in maintaining the structural integrity of the aortic wall. 

Despite diagnostic advances, the detection of TAAs in their early stages is difficult, since TAA is usually asymptomatic until aortic dissection or rupture occurs [6], which is associated with a mortality of approximately 50% [7]. Dilatation of the aorta increases the risk of dissection; however, the effectiveness of current medical therapies to slow the rate of aneurysmal growth is limited, with moderate-risk prophylactic surgery (mortality 1–5%) recommended at diameters >5 cm [8]. Importantly, dissection frequently occurs in TAAs at diameters well below this threshold [9], resulting in emergency surgery, which carries a mortality risk of approximately 12% [10], highlighting the need for early detection of patients who are at an increased risk of aortic dissection, irrespective of aortic diameter. Early detection is ultimately hindered by the clinical heterogeneity of TAA presentation, with considerable variability seen in the age of onset, penetrance, and progression of disease [11]. Additionally, the high degree of intra-familial variability in individuals who harbor the same causative pathogenic variant suggests that disease modifiers lying outside the causative variant are likely responsible for the phenotypic variability seen in these patients [12,13].

In order to identify potential disease modifiers, a well-understood pathogenesis of disease is needed. The role of inflammation and oxidative stress is well established in the pathogenesis of AAA [14]; however, it is more controversial in genetically triggered TAAs, which are traditionally classified as non-inflammatory lesions [15]. More recently, studies throughout the literature support the idea that inflammation and the subsequent formation of reactive oxygen species (ROS) may play a more important role in the pathogenesis of genetically triggered TAA than first thought. This review will explore the emerging evidence of inflammation and oxidative stress in the pathogenesis of genetically triggered TAAs, with a focus on links between inflammation, myeloperoxidase (MPO) production, and its consequences in the vasculature, to examine how these factors may contribute to TAA disease progression.

## 2. Pathogenesis of Thoracic Aortic Aneurysms

### 2.1. Structure of the Thoracic Aorta and Its Extracellular Matrix Components

The thoracic aorta is composed of an arrangement of endothelial cells (ECs), VSMCs, ECM proteins, and fibroblasts organized into three distinct layers: tunica intima, tunica media, and tunica adventitia. The tunica intima consists of a thin layer of ECs lining the lumen of the vessel, laying upon a basement membrane made up of a highly specialized ECM network, with the main components including laminin, collagen type IV, fibronectin, perlecan, and heparan sulphate proteoglycans. The tunica media consists of approximately 60 lamellae, consisting of elastin interspersed with VSMCs, embedded in extracellular-associated collagens (mainly type III) and glycosaminoglycans, which together form the contractile-elastic unit, whose function is to both bear and sense tension. The tunica adventitia is rich in collagen (mainly type I) and fibroblasts and contains the local blood (vasa vasorum) and neural supply. Its purpose is to provide tensile strength to the aortic wall [16].

### 2.2. Histopathology—Medial Degeneration

The thoracic aorta plays a pivotal role in the regulation and maintenance of ventricular performance as well as maintaining constant blood flow throughout the entire cardiovascular system [17]. The elastic properties of the medial layer of the aortic wall allow energy from systole to be stored and recovered in diastole through elastic recoil, allowing blood to continue in a forward direction while maintaining pulse pressure [18]. These mechanical properties of the aortic wall are maintained through the highly organized arrangement of VSMC, layers of elastic fibers, and ECM proteins [6,19]. Disturbances to this highly organized structure of the aortic wall will therefore impede the mechanical function of the vessel. 

During TAA formation, changes to these components result in specific histopathological features of the aortic wall, described as medial degeneration. The most prominent feature is the fragmentation, disorganization, and loss of elastic fibers, which reduces aortic compliance and recoil, therefore leaving the aorta prone to dilation and dissection [18]. Elastic fiber degeneration, in conjunction with loss of VSMCs, results in cystic-like spaces within the media, which fill with a collection of basophilic-mucopolysaccharides, another hallmark of medial degeneration. This focal pooling contributes to forcing the lamellar sheets apart, thus separating elastin from its supporting collagen, leading to increased risk of aortic dissection [20]. Particularly in the case of genetically triggered TAA, very few inflammatory cells are observed within the aortic wall [15].

### 2.3. Molecular Mechanisms of Pathogenesis

Although the histological characteristics of TAA are well characterized, the processes that lead to these changes within the aortic wall are only partly understood. It is likely that several mechanisms contribute to medial degeneration, all affecting aortic wall homeostasis and favoring excessive ECM degradation, accumulation of proteoglycans, and loss of VSMCs [21]. These mechanisms include abnormal mechanosensing of aortic wall stress, dysregulation of TGF-β signaling, and VSMC phenotype switching.

Homeostasis of the normal aortic wall is maintained through mechanosensing mechanisms, where the connections between the ECM and VSMCs are able to sense and communicate aortic stretch, allowing for correct remodeling of the aortic wall to occur in response to stimuli. Dysfunctional mechanosensing of the aortic wall has been shown to contribute to the formation of TAAs [22]. The normal mechanisms in homeostasis in response to appropriate mechanotransduction within the aortic wall, and how pathogenic variants associated with genetically triggered TAAs disrupt this mechanosensory feedback loop, have been previously reviewed in detail [23].

Increases in both TGF-β and angiotensin-II signaling occur as a consequence of defective mechanosensing by aortic wall VSMCs, which results in increased expression of matrix metalloproteinases (MMPs), a family of proteinases that play a role in degrading components of the ECM, including collagen and elastin [16]. VSMC phenotype switching from a contractile phenotype to a synthetic active phenotype is also seen in TAA [23]. The synthetic VSMC phenotype is characterized by the secretion of ECM proteins and MMPs, consistent with the idea of altered aortic wall homeostasis in TAA pathology. Additional contributing factors, aside from the causative pathogenic variant, may exacerbate these pathological mechanisms and play a role in modulating the severity of TAA progression. 

## 3. Genetically Triggered Thoracic Aortic Aneurysms

### 3.1. Marfan Syndrome

Marfan syndrome (MFS) is an autosomal dominant inherited connective tissue disorder characterized by abnormalities in the eye (primarily ectopia lentis), disproportionate overgrowth of the skeleton, and cardiovascular abnormalities. Aortic dilation, particularly of the aortic root sinuses, is the most life-threatening feature of MFS [24] and is detectable in 90% of patients over the age of 60 years [25].

MFS is consequent upon pathogenic variants in *FBN1*, which encodes fibrillin-1, an essential component of extracellular matrix microfibrils. Microfibrils are closely associated with the elastin lamellae [26]. It is thought that the pathogenesis of MFS-TAA may involve two mechanisms: structural impairment to the aortic wall caused by abnormal fibrillin-1 production and aberrant signaling by the TGF-β pathway, as fibrillin plays a role in regulating the bioavailability of TGF-β [27]. Upregulation of the TGF-β pathway is thought to lead to pathological remodeling of the aortic wall through upregulation of ECM degradation proteins, particularly MMPs, resulting in the characteristic medial degeneration described above, which is observed in MFS-TAA tissue specimens. 

### 3.2. Loeys–Dietz Syndrome

Loeys–Dietz syndrome (LDS), like MFS, is an inherited connective tissue disorder with aneurysm formation being the most life-threatening complication. Diagnosis is usually based on the classic triad of arterial tortuosity, hypertelorism, and wide or split uvula. However, LDS patients exhibit more widespread aneurysm formation, including involvement of the aortic root, arterial branches of the head and neck, lung, and lower extremities [28]. Patients also experience skeletal, cutaneous, and ocular abnormalities, including scoliosis and pectus deformities, and velvety thin translucent skin. 

The pathogenic variants responsible for LDS are genes that encode proteins within the TGF-β signaling pathway and include the TGF-β ligands (*TGFB2* and *TGFB3*), their receptors (*TGFBR1* and *TGFBR2*), and their intracellular signaling mediators (*SMAD2/3*) [29].

### 3.3. Vascular Ehlers–Danlos Syndrome

The vascular type of Ehlers–Danlos syndrome (vEDS) is a rare subtype of EDS [30], with a prevalence of approximately 1:200,000. vEDS is characterized by generalized tissue fragility, including within the vasculature, and is caused by pathological variants in *COL3A1*, which encodes the type III collagen alpha-chain [31]. Widespread arterial rupture and dissection is the leading cause of premature death at a median age of 50 years [30].

### 3.4. Bicuspid Aortic Valve 

Bicuspid aortic valve (BAV) is the most common congenital heart disease, with an estimated prevalence of 0.4–2.25% [32]. Abnormal aortic cusp formation during development leads to two abnormal, rather than three normal, aortic cusps. BAV has an increased prevalence in Caucasian cohorts as well as a male to female prevalence ratio of approximately 3:1 [33]. Up to 50% of adult BAV patients show evidence of TAA dilatation [34].

Two potentially overlapping theories to explain TAA formation in BAV patients have been extensively debated: the “hemodynamic” and the “genetic” theories. The hemodynamic theory suggests that the malformed BAV will result in turbulent blood flow, increasing the stress on the aortic wall and predisposing these individuals to TAA development [35], while the genetic theory suggests that there is a genetically triggered intrinsic weakness within the aortic wall of BAV patients, irrespective of the level of valvular dysfunction [36,37]. It is likely that there is a complex interplay between both altered hemodynamic flow and genetic factors that contribute to TAA formation in BAV patients. Irrespective of the cause, the risk of aortic dissection in BAV patients is 9-fold higher than the risk in the general population [33].

### 3.5. Familial Thoracic Aortic Aneurysm and Dissection

Non-syndromal forms of TAA may be familial or sporadic. Familial thoracic aortic aneurysm and dissection (fTAAD) is an inherited predisposition to aneurysm formation occurring independently from an underlying condition or syndrome and the diagnosis requires either a family history and/or the identification of a pathogenic variant within the family. fTAAD has been estimated to occur in greater than 20% of families with non-syndromal TAA, but this figure may be substantially higher [38]. fTAAD patients have a relatively early age of onset, a higher aortic growth rate (0.21 cm/year), and a higher incidence of fatal aortic dissection, compared to sporadic cases of non-syndromal TAA [38], highlighting the aggressive nature of fTAAD [39]. Additionally, fTAAD males are at a higher risk of aortic dissection than affected females [40]. The etiology of fTAAD is largely unknown, although in approximately 30% of families with fTAAD, a pathogenic variant can be identified [3]. Pathogenic variants in genes causing fTAAD may be involved in a diverse array of biological functions, including contractility of VSMCs (*ACTA2, MYH11, MYLK, PRKG1*), TGF-β signaling (*TGFBR1, TGFBR2, TGFB2*, and *SMAD3*) [41], and maintenance of ECM (*LOX*) [3].

### 3.6. Sporadic and Degenerative Thoracic Aortic Aneurysms

Sporadic TAA is also considered non-syndromal and represents those patients in which no family history related to TAA can be elicited, and in whom genetic testing fails to detect a pathogenic genetic variant [42]. On the other hand, degenerative TAA usually occurs in older patients who exhibit a history of environmental risk factors, such as hypertension, smoking, and atherosclerotic vascular disease. In both sporadic and degenerative TAA, it is likely that many of these patients harbor a genetic predisposition for TAA formation, where the genetic predisposition may be a variant in a single or multiple genes, and where the variant/s may be, by themselves, sub-clinical, requiring additional environmental factors to produce a pathological phenotype [43]. Notably, a genome-wide association study comparing 765 subjects with sporadic TAA with 874 controls found associations with the chromosome 15q21.1 locus that includes the *FBN1* gene, providing a potential link between the pathogenesis of MFS and sporadic TAA [43,44].

## 4. Evidence of Inflammation in Thoracic Aortic Aneurysms

Inflammation within the vasculature is an essential feature in the pathogenesis of numerous diseases, including atherosclerosis and AAA. While the role of inflammation in genetically triggered TAAs is less established, there is growing evidence implicating the inflammatory response in TAA pathogenesis and the progression of disease. Notably, inflammation may occur early in the disease, thus being a driving factor, and/or later, as a consequence of the disease, where the response is a mal-adaptive damaging compensatory process. A limitation of clinical studies of thoracic aortic tissue is the availability for harvest of only end-stage surgical tissue, rendering the chronology of the inflammatory process difficult to assess.

Studies of genome-wide gene expression and DNA methylation in TAA samples have revealed changes in genes and pathways pivotal to the immune response. Gene expression analysis performed on 32 degenerative ascending TAA tissue samples showed enrichment of differentially expressed genes involved in inflammatory pathways, when compared to healthy aortic wall specimens [45]. A similar finding was observed in our own study, which analyzed the genome-wide DNA methylation status in peripheral WBCs of MFS patients and compared these to control samples [46]. Using gene ontology analysis, enrichment of inflammatory genes was observed in genes that were differentially methylated between patients and controls, including those involved in positive regulation of interferon-gamma production and the interferon-gamma-mediated signaling pathway [46]. Supporting our data, ascending TAA tissue, predominantly from patients suffering from degenerative TAA, has been shown to have increased expression of interferon-gamma when compared to control aortic tissue [47]. These studies indicate that the regulation of inflammatory processes is altered in several forms of TAA, suggesting inflammation has a role in the pathogenesis of TAA.

### 4.1. Monocytic/Macrophage Infiltration

Moderate inflammatory infiltrates of the aortic wall are a well-characterized feature of established mouse models of MFS. The fibrillin-1 hypomorphic mouse model (mgR/mgR) [48,49] displays a monocytic infiltration of the medial layer. Additionally, expression of monocyte chemotactic protein-1 (MCP-1), a key chemokine in regulating the migration and infiltration of monocytes/macrophages, was observed, which correlated with the extent of elastin fragmentation and adventitial inflammation in these mice [48,50]. Similarly, increased MCP-1 expression was found to correlate with more severe aortic dilatation in another MFS model mouse [51]. Additionally, monocytic chemotaxis in the *Fbn1* mgR/mgR mouse model of MFS has been shown to be mediated by elastin fragments, released following proteolytic degradation, probably mediated by synthetic VSMC- and macrophage-derived proteases, including MMPs [49]. Thus, the characteristic elastin fragmentation seen in MFS likely initiates and perpetuates the infiltration of inflammatory cells into the aortic wall, allowing a destructive cycle to ensue. The increased chemotactic properties of aneurysmal aortic extracts have been validated in human MFS TAA tissue, strongly implicating elastin degradation products as drivers of inflammation in MFS [52]. 

SMAD4 is the key Co-SMAD in the canonical TGF-β signal transduction cascade. Loss-of-function SMAD4 mutations result in TAA formation in humans [53] and exacerbate aortic disease in a MFS mouse model [54]. Additionally, a variant SNP (rs12455792) located in 5′UTR of the *SMAD4* gene, which leads to lower SMAD4 expression in sporadic TAA tissue, correlates with an elevated risk of TAA and dissection [55]. Furthermore, low SMAD4 expression has been shown to facilitate inflammatory cell infiltration and accumulation in both human TAA tissue [56] and in SMAD4-deficient mice, who develop TAAs associated with abundant leukocyte infiltration, predominantly macrophages, within the aortic media [57]. Thus, SMAD4-mediated TGF-β signal transduction appears to be a key element in reducing inflammatory cell activation, while reduced SMAD4 expression results in more severe TAA disease.

Increased macrophage infiltration has been observed in the media of aortic tissue of both MFS patients [49,58] and fTAAD/sporadic TAAs [58,59], although the infiltrate was sparser in sporadic cases, indicating that inflammation may be more crucial to the pathogenesis of TAAs in which a known genetic trigger has been demonstrated. Taken together, these data strongly support a role for macrophage recruitment in TAA pathogenesis, highlighting the involvement of the inflammatory response.

### 4.2. T-cells

Increased T-cell activation has been demonstrated in MFS, fTAAD, and sporadic TAAs, with CD3^+^ T cells shown to be present in the aortic media and adventitia of aneurysmal tissue [58,59]. More specifically, in MFS, the TAA media contains predominantly CD4^+^ T-helper cells, whose activities include the regulation of fibrotic responses, while the adventitia contains CD8^+^ cytotoxic T cells [60]. In atherosclerosis, CD8^+^ T cells have been shown to secrete interferon-gamma, which has the capacity to induce apoptosis of VSMCs in vivo [61], which is also a hallmark of TAA pathogenesis. 

Additionally, a genome-wide analysis of BAV TAA tissue has demonstrated both hypermethylation and underexpression of the protein tyrosine phosphatase non-receptor type 22 (*PTPN22*) gene when compared to aneurysmal tissue from patients with a tricuspid aortic valve [62]. Thus, increased T-cell signaling will result from the decrease in the expression of the protein coded by *PTPN22*, lymphoid tyrosine phosphatase, which is a key regulator of T-cell signaling, consistent with an increase in T-cell activity in these BAV TAA patients. 

### 4.3. Interleukins 

Several interleukins have been implicated in TAA development, including IL-6 [50,63,64], IL-10 [65,66], and IL-1β [67,68,69,70]. IL-6 is a hallmark of vascular inflammation [71], with several IL-6-KO mouse models demonstrating that IL-6 deficiency mitigates TAA pathogenesis, in both a TAA elastase treatment model [64] and the *Fbn1* mgR/mgR mouse model of MFS [50]. IL-6 deficiency led to a decrease in TAA size, but in the case of the *Fbn1* mgR/mgR•IL-6^−/−^ double mutant mouse model, IL-6 deficiency did not alter the rate of rupture and consequently did not improve survival. An increase in IL-6 levels during the pathogenesis of human TAA has been demonstrated in both plasma [63] and TAA tissue [64], in patients suffering from the degenerative form of TAA. Importantly, a recent study focusing on genetically triggered TAA has shown a strong correlation between circulating IL-6 and increased aortic dimension in all segments of the aorta [72]. One of the most potent cardiovascular inducers of IL-6 is angiotensin II (AngII), with the activation of the angiotensin type 1 receptor causing VSMC IL-6 expression [73]. AngII signaling is implicated in the pathogenesis of MFS TAA formation, with AngII receptor blockers being tested as a potential drug therapy to reduce aortic dilatation, however, with mixed results [74]. While the interaction between the AngII system and TGF-β signaling has been studied in MFS TAA pathogenesis, the increase in IL-6 seen in TAA tissue warrants investigation into whether this increase is a consequence of aberrant AngII signaling known to be involved in TAA pathogenesis. 

As an anti-inflammatory cytokine, IL-10 would be expected to mitigate inflammatory damage occurring within the aortic wall [75]. While IL-10 mRNA levels are decreased in the aortic arch of the *Fbn1*+/1037G MFS mouse model [65], IL-10 mRNA has been found to be overexpressed in peripheral blood leucocytes from degenerative TAA patients [66]. IL-10 is predominately secreted by macrophages and CD4+ helper T-cells, both of which infiltrate TAA tissue [49,52,59,60], and is thought to help limit the potential damaging products generated by macrophages, including MMPs and proinflammatory cytokines [76]. Thus, an increase in IL-10 expression may represent a compensatory mechanism to limit further tissue injury. These results indicate that further investigation is necessary to characterize the role of IL-10 in the pathogenesis of TAA.

IL-1β is a strongly proinflammatory cytokine secreted by macrophages [77]. Genetic deletion of IL-1β in a mouse model of TAA has been shown to significantly decrease thoracic aortic dilatation, while mouse models of aortic dissection have demonstrated increased levels of IL-1β in aneurysmal tissue, highlighting a potential role of IL-1β expression in TAA development [67,68,69]. In human thoracic aortic tissue samples from patients presenting with TAA or dissection, IL-1β levels have been shown to be significantly elevated when compared to control aortic tissue [68,70]. Interestingly, IL-1β levels were shown to be significantly higher in tissue collected from patients undergoing elective TAA repair when compared to tissue collected from patients undergoing emergency repair for thoracic aortic dissection, suggesting that IL-1β plays a more important role in modulating TAA growth rather than aortic dissection [70].

Limited data on the involvement of other interleukins in TAA formation and aortic dissection have also been obtained, with levels of IL-11, IL-17, and IL-22, all of which display some proinflammatory activity, shown to be significantly elevated in aortic tissue and in the circulation of TAA patients presenting with an acute dissection [78,79]. Additionally, upregulation of the proinflammatory cytokine IL-3 signaling pathways and an increase in IL-3-positive cells in aneurysmal tissue has also been demonstrated in a mouse model of TAA dissection [80]. This limited evidence supports the role of IL-11, IL-22, IL-17, and IL-3 in aortic dissection; however, further investigations are needed to determine whether these interleukins have a role in the early stages of TAA development.

### 4.4. Inflammation and Disease Progression

Several studies have implicated the extent of inflammation as a marker of the rate of TAA progression in MFS. MFS patients were stratified into progressive (0.9 mm/year) or low (0.1 mm/year) aortic dilatation rate groups, and an RNA expression microarray of skin biopsies from these patients was undertaken. Patients with MFS exhibit abnormal connective tissue, including within the skin [28]. The skin biopsies revealed increased expression of *HLA-DRB1* and *HLA-DRB5* within the progressive group [60]. These genes code for the heavy chain of major histocompatibility complex II, which is important in the activation of T helper cells, suggesting an increased inflammatory response in patients with a faster rate of aortic dilatation. Additionally, an increase in circulating plasma macrophage-colony stimulating factor (M-CSF), a strong macrophage attractant, was observed in the progressive group in the same study, although the 10-fold elevated level of TGF-β seen within the MFS group overall did not correlate with the severity of the progression of TAA disease, as measured by the aortic diameter or aortic dilation rate [60]. Thus, while these results suggest TGF-β signaling is important in initiating aortic dilatation, the extent of the inflammatory response seems to be involved in the progression of disease severity. Taken together, these data are consistent with the hypothesis that inflammation may play a more substantial role in genetically triggered TAA than was initially thought, justifying ongoing investigation.

## 5. Inflammation and Oxidative Stress: Myeloperoxidase and Reactive Oxygen Species

Reactive oxygen species (ROS) are defined as small oxygen-derived molecules, which readily react with other molecules. While the balance between ROS production and ROS removal is essential for normal cell function, excess ROS production leads to oxidative stress and detrimental damage to RNA, DNA, proteins, and lipids, implicating ROS and oxidative stress in the pathogenesis of many diseases [81].

Most intracellular ROS production is derived from the mitochondria, as a by-product of mitochondrial electron transport during aerobic respiration [82]. Another major source of ROS production is enzymatic activity of NADPH oxidases (NOX), in which ROS production is the main function of the enzymatic system rather than a by-product of normal enzymatic activity [83]. Seven NOX family homologs, found in a variety of tissues, have been discovered (NOX1-NOX5, DUOX1, and DUOX2), all being transmembrane proteins that transport electrons across biological membranes to reduce oxygen molecules to superoxide (O_2−_).

Specifically, phagocyte NOX2 has a role in catalyzing the respiratory burst that occurs during phagocytosis to assist in the inactivation and removal of invading pathogens. Activation of the phagocyte causes an increase in oxygen consumption, with NOX2 facilitating the reduction of oxygen, using NADPH as an electron donor, resulting in the formation of O_2−_ [83]. O_2−_ is then rapidly converted to the less reactive ROS, hydrogen peroxide (H_2_O_2_), either spontaneously or by the enzyme superoxide dismutase [84].

### 5.1. Myeloperoxidase

The combination of H_2_O_2,_ produced during the respiratory burst of phagocytosis, with the enzyme myeloperoxidase (MPO), significantly enhances the bactericidal activity of phagocytic cells through the generation of oxidizing species, such as the hypohalous acids [85]. MPO is a hem-containing peroxidase, released from cytoplasmic granules of phagocytes, and specifically makes up 5% of the dry mass of neutrophils, making it this cell’s most abundant protein [86]. MPO is a highly cationic dimeric protein with a mass of approximately 146 kDa. The *MPO* gene is located on the long arm of chromosome 17 (q23.1) and is highly conserved during evolution [87,88]. The synthesis of the mature MPO protein is a complex process, involving cleavage of the initial 90 kDa transcription product, incorporation of heme to produce an enzymatically active protein, followed by numerous proteolytic modifications [89]. Mature MPO consists of two identical monomers linked by a disulphide bridge, each consisting of a heavy-chain and light-chain subunit. The heavy chain is glycosylated and contains a protoporphyrin IX group with a central iron ion, located at the bottom of a deep crevice [90], limiting the accessibility of the iron to most materials except for H_2_O_2_ and other small anions. Oxidation of other materials by MPO occurs through binding to a hydrophobic pocket at the entrance to the distal heme cavity [91]. 

### 5.2. MPO-Derived Oxidants

MPO contributes to microbial elimination through the generation of oxidant species, with hypohalous acids, such as hypochlorous acid (HOCL) and hypobromous acid (HOBr), being the main oxidants responsible for the bactericidal activity of phagocytes [92]. The native ferric (Fe(III)) heme of MPO reacts with H_2_O_2_, which is generated during the respiratory burst of activated phagocytes, to form a reactive intermediate compound (compound I) via two-electron oxidation [93]. This compound, with its high oxidation state, can catalyze the conversion of halide and pseudohalide ions (Cl^−^, Br^−^, SCN^−^) to their corresponding hypohalous acids (HOCl; HOBr; and hypothiocyanous acid, HOSCN), via a two-electron reduction that converts compound I back to its ferric native state [91]. These reactions make up the halogenation cycle (native enzyme → Compound I → native enzyme). As Cl^−^ is the most abundant substrate for the halogenation cycle in plasma under physiological conditions, HOCl is the major species that is formed from the halogenation cycle [94]. HOCl is highly reactive, participating in both oxidation and chlorination of many biological targets. HOBr generation accounts for approximately 25% of the H_2_O_2_ that is consumed during the halogenation cycle [94,95].

A second competing enzymatic cycle, termed the peroxidase cycle, involves one-electron oxidation of various organic and inorganic substrates to form radical species. Compound I can undergo a one-electron reaction to generate compound II and free radicals [91]. Compound II is then reduced back to its native form in a second one-electron reduction reaction. One radical species that is formed as a result of the peroxidase cycle is nitrogen dioxide (NO_2_.), a product of NO_2_^−^ oxidation [88]. NO_2_. is a highly unstable molecule and promotes protein nitration and lipid peroxidation [96]. The modifications of these MPO-derived oxidants, such as HOCl and NO_2_^−^, can alter protein activity and therefore influence overall cellular functions, which can be beneficial or detrimental, depending on the circumstances. The generation of all MPO-derived oxidants has previously been reviewed in depth [97].

### 5.3. MPO in Disease

Oxidant production by MPO is an advantageous event during phagocytosis and the immune response, but substantial evidence has shown that excess levels of these toxic molecules cause tissue damage, with MPO-mediated damage being implicated in a wide range of inflammatory disease models, including cardiovascular and kidney disease, pulmonary and skin inflammation, neurological disorders, and metabolic syndrome, recently summarized in a review [85]. Of specific relevance is that elevated MPO levels predict future risk of coronary artery disease in healthy individuals [98].

Neutrophils are the main source of MPO, but it is also expressed at lower levels in circulating monocytes. While expression is generally lost in mature tissue macrophages, evidence suggests MPO expression can be induced in these cells under certain conditions, such as within microglia (macrophages of the brain) surrounding brain lesions in multiple sclerosis [99,100] and within macrophages within atherosclerotic lesions [101,102]. Additionally, MPO-expressing macrophages have been implicated in the pathogenesis of acute coronary syndrome [103,104,105] and SNPs located in the MPO promoter (the MPO-463G/A and -129G/A polymorphisms) have been shown to correlate with coronary artery disease [106].

In pathological conditions, MPO has also been detected in cells outside the myeloid lineage, including endothelial cells [103,107], neurons [108,109], prostate tissue [110], and astrocytes [111]. Additionally, endocardial endothelial expression of MPO has also been observed following oxidative stress, both in tissue culture and in post-infarcted human heart tissue [103].

## 6. MPO-Associated Oxidative Stress and Links to TAA Pathogenesis

The data discussed above support the hypothesis that macrophages that are capable of producing MPO can infiltrate into TAAs and are likely involved in TAA disease progression. There have been several recent studies that have directly implicated MPO activity in the pathogenesis of aneurysm formation, and, specifically, limited animal and human data for direct involvement of MPO in TAA. Additionally, indirect evidence from a number of studies have demonstrated that MPO can exacerbate mechanisms that are known to be involved in TAA pathogenesis, such as damage to the ECM and DNA, activation of inflammatory signaling, and an increase in endothelial dysfunction.

### 6.1. The Presence of MPO in TAA Samples and Other Aneurysmal Tissue

Increased MPO expression has been observed in a range of different types of aneurysmal tissue, providing evidence that increased expression of MPO may also promote TAA formation. Increased MPO expression has been demonstrated within locally sampled plasma and within the walls of human cerebral aneurysms, and reduced MPO expression was found to correlate with decreased cerebral aneurysm formation and rupture in an MPO-knockout mouse model [112]. Additionally, in human cerebral aneurysm samples, MPO has been shown to be associated with a neutrophil inflammatory cell infiltrate, degenerative remodeling of the arterial wall, and wall rupture [113]. Notably, cerebral aneurysms can accompany TAA associated with *TGFBR* and *ACTA2* mutations [41,114].

In AAA, MPO has been shown to be important in both the initiation and progression of disease. *MPO* gene deletion in a mouse model of AAA prevents aneurysm formation [115] and higher levels of MPO and H_2_O_2_ in circulating neutrophils are observed in AAA patients [116]. Higher levels of MPO have also been associated with faster AAA growth in humans [117,118].

Increased MPO has been observed in the aortic wall of a mixed cohort of patients requiring surgical resection for TAA, with MPO expressed within the intimal and medial layers [119]. Additionally, in a small study of BAV patients, plasma MPO has been specifically associated with valvular dysfunction, but no effect was seen on TAA development or progression in this underpowered study [120].

A mouse model of MFS (*Fbn1* C1039G/+ mice) provides strong evidence of the involvement of MPO in the pathogenesis of TAA [121]. MPO knockout in this MFS mouse model resulted in decreased ROS production, associated with decreased VSMC apoptosis, decreased elastin fragmentation, and decreased MMP activation within the aortic wall, consistent with a central role for MPO in mediating these important pathological characteristics of MFS aneurysms.

### 6.2. MPO and the Extracellular Matrix 

Disturbed ECM homeostasis is characteristic of TAA formation. The ECM is susceptible to oxidative damage; lymphocyte-derived oxidants, including the MPO-derived oxidants HOCl and HOBr, can attack ECM proteins, including proteoglycans, fibronectin, and laminin, inducing chemical and structural changes to these proteins. 

Cultured VSMCs from MFS and BAV patients have been shown to exhibit differences in the localization and distribution of fibronectin, although the total amount of fibronectin remained unchanged between controls and diseased cells [122]. In human coronary artery VSMCs, both HOCl and an MPO enzymatic system can generate site-specific alterations to fibronectin, leading to functional abnormalities that affect adherence, proliferation, and expression of ECM synthesis and remodeling genes [123]. The changes in fibronectin distribution in cultured VSMCs from MFS and BAV patients [122] may in part be mediated via MPO-derived oxidants.

Similarly, human atherosclerotic lesions exhibit extensive and site-specific modifications of the key basement membrane protein laminin, by MPO-derived oxidants, including within the integrin binding region, which likely leads to abnormalities in cellular adhesion and cell–matrix interactions [124]. Notably, alterations in the integrin binding region may contribute to the mechanosensory detection issues that are present in TAA pathology [23].

Exposure of human coronary artery VSMCs to HOCl has revealed alterations to the structures of fibronectin, type IV collagen, laminin, and versican, which led to reduced cell adhesion, increased cell proliferation, and changes in VSMC gene expression, including increased expression of inflammatory genes and MMPs, consistent with the VSMC phenotypic changes often seen in vascular injury [125]. Thus, the phenotypic switching seen in TAA [23] may be mediated in part by oxidative stress-induced alterations to ECM proteins. 

### 6.3. MPO and Matrix Metalloproteinase Activation

The balance between activation and inactivation of MMPs during proteolytic events is tightly regulated, as uncontrolled activity can be destructive and contribute to various pathologies, including TAA formation [126,127]. To assist with this balance, MMPs are first synthesized as inactive pro-enzymes, which contain a “cysteine switch” motif sequence, where a free cysteine residue interacts with the catalytic zinc ion to keep the pro-enzyme in a latent state [126]. Therefore, activation of the pro-MMP requires removal of the cysteine switch or a conformational change that will break the interaction of the domain with the zinc in the active site, leaving it available to interact with the relevant substrates.

One mechanism known to alter the cysteine switch involves a chemical modification of the free cysteine residue by ROS. Studies in vitro show that MPO-derived HOCl may be an oxidant that can regulate MMP activation [128,129,130,131,132,133]. The potential mechanism for this regulation involves HOCl oxygenating the thiol residue of the cysteine switch domain, converting the latent MMP to its active form [128]. 

In vitro studies of HOCl-dependent activation of pro-MMPs demonstrate activation at moderate concentrations of HOCl but inactivation at higher concentrations [131], with activation being dependent on the type of MMP. Collagenases (MMP-1, 8, 13, and 18) are activated in response to HOCl, while gelatinases (MMP-2 and 9) show little HOCl-dependent activation but are more sensitive to HOCl-dependent inactivation. Another study has demonstrated that MPO-derived HOCl can directly inhibit MMP activity, since exposing the matrilysin MMP-7 to increasing concentrations of HOCl converted it to an enzymatically inactive form of the protein through alterations of the active domain of the enzyme [129]. Contradictory results have been seen in stimulated human neutrophils, where the addition of an MPO inhibitor (azide) or an HOCl scavenger (methionine) both significantly reduced the amount of active MMP-9 in neutrophil supernatants, without affecting the amount of MMP-9 released following neutrophil stimulation [130].

Additionally, an in vivo mouse model of neuroinflammation has shown that MMP activity is reduced in the brain when MPO is inhibited [134]. MPO activity and MMP-9 activity were shown to be strongly correlated with each other in brain extracts from these mice, suggesting a higher level of complexity in vivo.

Another important mechanism responsible for regulating the level of MMP activation is through the activity of the endogenous inhibitory proteins, tissue inhibitors of metalloproteinases (TIMPs). MPO-derived HOCl has been shown to inactivate TIMP-1 by oxidizing its N-terminal cysteine in vitro, dramatically decreasing its ability to inactivate MMPs, with confirmation of the clinical relevance of the production of this modification being found in the bronchoalveolar lavage fluid from patients suffering from acute respiratory distress syndrome [135].

### 6.4. MPO and Activation of the ERK1/2 Signaling Pathway

Increased activation of non-canonical ERK1/2 signaling is known to be associated with TAA pathogenesis, predominately through the downstream effect of increased MMP expression. These signaling pathways are known to be redox sensitive, with increased oxidative stress due to inflammation, including the production of MPO-derived oxidants, leading to increased activation [136]. Both treatment of endothelial cells with MPO [107] and VSMC with HOCl has been shown to result in increased phosphorylation of ERK1/2 [137]. Thus, increased MPO can also indirectly increase MMP expression through ERK1/2 signaling.

In addition to increasing MMP expression, the activation of ERK1/2 signaling pathways through redox-related events can disrupt aortic wall homeostasis by altering the characteristics of VSMCs. Activation of ERK1/2 in the presence of both 3-nitrotyrosine and 3-chlorotyrosine, markers of MPO-mediated oxidative damage, has been shown to promote human aortic VSMC migration [138,139], consistent with the pathological synthetic phenotype associated with TAA pathogenesis [140]. Nitrotyrosine levels have been shown to be increased in the aortic medial layer of both aggressive forms of human TAA [141] and human TAA tissue and cultured smooth muscle cells from MFS patients [141,142]. Thus, the increase in nitrotyrosine seen in TAA may be contributing to the VSMC dysfunction involved in the pathogenesis of TAA through an increase of ERK1/2 signaling. 

The mechanisms by which MPO and its derived oxidants activate ERK1/2 signaling pathways are not well understood [136]. Kinase phosphorylation is required to activate MAPK signaling pathway cascades, which leads to expression of relevant target genes, including upregulation of MMPs. There is evidence to suggest that MPO-derived oxidants can oxidize thiol groups of protein tyrosine phosphatases, which are responsible for the dephosphorylation of MAPK signaling proteins, causing a significant loss in enzyme activity, therefore directly affecting the phosphorylation cascade and causing overactivation of the ERK1/2 signaling pathways [143,144].

### 6.5. MPO, Nitric Oxide Availability, and Endothelial Dysfunction

Nitric oxide (NO) is an important regulator of vascular tone, with loss of NO bioavailability implicated in endothelial dysfunction [145]. MPO has the ability to reduce NO bioavailability through direct oxidization of NO or through the production of NO-consuming substrate radicals, thus contributing to the endothelial dysfunction seen in many vascular pathologies [146,147], including TAA.

Impaired endothelial function as a result of reduced NO production has been observed in the thoracic aorta of MFS mice [148], although this study did not directly examine a link to MPO activity. These data support the hypothesis that increased MPO activity may, in part, mediate reduced NO bioavailability, thus contributing to intrinsic endothelial dysfunction in MFS and worsening TAA phenotype. 

### 6.6. MPO Utilizing Vascular NADPH Oxidase-Derived H_2_O_2_


The expression of NOXs has been investigated in MFS aortic tissue and cultured VSMCs, as NOXs are the main source of ROS in the cardiovascular system [142]. NOX4 expression was seen to be increased in MFS tissue and VSMCs when compared to controls, which also correlated with an increase in H_2_O_2_ in MFS tissue. NOX4 expression is known to be upregulated by TGF-β, which was supported by a decrease in NOX4 expression when the TGF-β receptor was inhibited. An MFS mice model deficient for NOX4 expression showed reduced elastin fragmentation, less endothelial dysfunction, and an increase in contractile markers, when compared to standard MFS mice. These data highlight the important role of NOX4 and its derived ROS species in the progression of aneurysm formation in an MFS mouse model [142].

While it is well established that MPO utilizes leukocyte-derived H_2_O_2_, it is less clear whether MPO uses vascular non-leukocyte oxidase-derived H_2_O_2_ to form its oxidants, including HOCl, to cause vascular injury. Vasculature NADPH oxidase-derived H_2_O_2_ has been shown to be utilized by MPO to produce HOCl, which was shown to amplify H_2_O_2_-induced vascular injury [149]. These findings suggest that under inflammatory conditions, MPO could exacerbate the vascular NADPH oxidase-derived ROS-induced vascular injury that has been demonstrated to occur during TAA pathogenesis.

### 6.7. MPO and Modifications to DNA

MPO-derived oxidants also lead to DNA modification, causing DNA damage. Specifically, MPO-derived HOCl has been linked directly to damage to specific DNA bases, and has been shown to inhibit DNA repair enzymes [88]. Additionally, HOCl-mediated DNA base oxidation has been observed in human aortic VSMCs treated with physiologically relevant levels of HOCl [150]. Furthermore, an increase in DNA damage has been observed in VSMCs isolated from human TAA tissue [151]. Thus, the possibility of MPO-derived oxidants affecting the DNA of VSMCs in the context of TAA warrants further investigation. 

## 7. Conclusions

Genetically triggered TAAs include a variety of syndromes and associated gene variants. Some forms of TAA have been clearly associated with a known primary pathogenic gene variant, such as *FBN1* in MFS, while the causative variants/s remain unknown for other forms of familial TAA, and BAV. Many of these genetically triggered TAAs exhibit only limited direct histological evidence of inflammation. However, closer evaluation of the pathogenesis of these TAAs has increasingly identified a role for inflammation at a subtle but significant level, which particularly involves an increase in inflammation-associated oxidative stress. Evidence is emerging of the likely contribution of MPO as a major contributor to oxidative stress in a number of disparate vascular pathologies, including genetically triggered TAA.

Figure 1 summarizes the proposed role of inflammation and MPO in the pathogenesis of genetically triggered TAA. We hypothesize that TAA development leads to an increased inflammatory infiltrate, mediated at least in part through increased chemotactic signaling from connective tissue fragmentation [49] and the expression of chemotactic cytokines, such as MCP-1 [48,50,51] and M-CSF [60]. The increased inflammatory infiltrate corresponds with an increase in MPO-derived oxidant generation. These oxidants directly damage the ECM, activate MMPs, both directly and through ERK1/2 signaling, and inactivate TIMPs. Additionally, these oxidants precipitate endothelial dysfunction and the switching of VSMCs to the pathological synthetic phenotype. All these pathological alterations lead to increased disruption of aortic wall homeostasis, resulting in a repeating cycle of more severe TAA development, which highlights the need for further investigations into the direct involvement of MPO in TAA.

Overall, the relationship between inflammation and genetically triggered TAA constitutes fertile ground for expanding our understanding of the pathogenesis of TAA, identifying potential biomarkers for early detection of TAA, monitoring severity and progression, and for defining potential novel therapeutic targets.

## Figures and Tables

**Figure 1 ijms-21-07678-f001:**
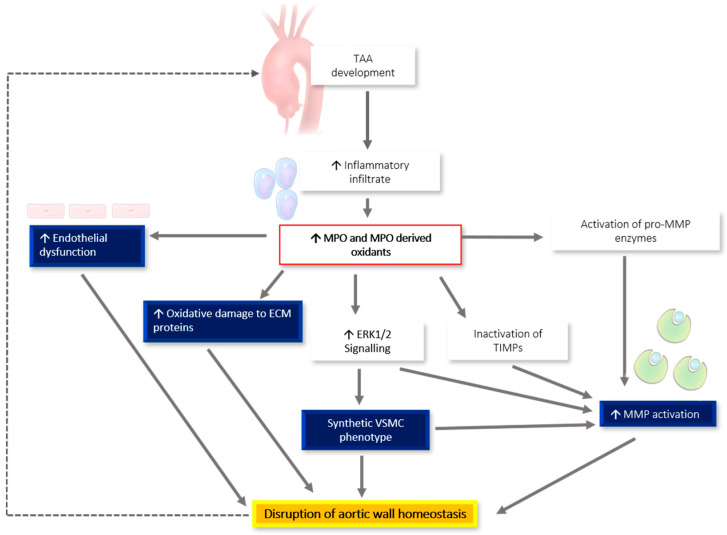
Proposed relationship between inflammation, MPO production, and worsening TAA phenotype. The development of a TAA leads to an increased inflammatory infiltrate and increased MPO-derived oxidant generation, which directly damage the ECM, activate MMPs, both directly and through ERK1/2 signaling, and inactivate tissue inhibitors of metalloproteinases (TIMPs). These oxidants also precipitate endothelial dysfunction and vascular smooth muscle cell (VSMC) phenotypic switching. Taken together, these changes disrupt aortic wall homeostasis, resulting in a repeating cycle of more severe TAA development.

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
