# Peer review of "The Role of Inflammation and Myeloperoxidase-Related Oxidative Stress in the Pathogenesis of Genetically Triggered Thoracic Aortic Aneurysms"

_ijms, 2020, doi:10.3390/ijms21207678_

Round 1

Reviewer 1 Report

This is a review article regarding pathogenesis of genetically triggered TAA. This reviewer considers that it was well written. This reviewer has some minor comments as described below.

Minor comment:

  1. Page 6. The authors indicated interleukin-6 and -10 as evidence of inflammation in TAA; however, there have been published regarding other interleukins in TAA field, based on PubMed. The author should mention regarding other interleukins.
  2. Page 12. At the end of conclusions section, the authors mentioned about novel therapeutic targets based on this review article, but this reviewer feels that it seems that the inflammatory responses in the genetically triggered TAA might be useful also for the development of novel biomarkers for early detection. If this is correct, the authors should mention not only treatment but also early detection.  

Author Response

Reviewer 1

This is a review article regarding pathogenesis of genetically triggered TAA. This reviewer considers that it was well written. This reviewer has some minor comments as described below.

  1. Page 6. The authors indicated interleukin-6 and -10 as evidence of inflammation in TAA; however, there have been published regarding other interleukins in TAA field, based on PubMed. The author should mention regarding other interleukins.

Response to reviewer: We agree that there are some additional papers that include evidence for the involvement of other interleukins in TAA pathogenesis. As suggested, we have included these papers in our review.

Specifically:

  • We have combined the sections on IL-6 and IL-10 into one section (Lines 276-279)

“Several interleukins have been implicated in TAA development including IL-6 [50,63,64], IL-10 [65,66], and IL-1β [67-70].”

  • We have introduced two new paragraphs to the manuscript that discuss the involvement of interleukins 1β, 3, 11, 17 and 22 in the pathogenesis of TAA (Lines 305-323)

“IL-1β is a strongly pro-inflammatory cytokine secreted by macrophages [77]. Genetic deletion of IL-1β in a mouse model of TAA has been shown to significantly decreased thoracic aortic dilatation, while mouse models of aortic dissection have demonstrated increased levels of IL-1β in aneurysmal tissue, highlighting a potential role of IL-1β expression in TAA development [67-69]. In human thoracic aortic tissue samples from patients presenting with TAA or dissection, IL-1β levels have been shown to be significantly elevated when compared to control aortic tissue [68,70]. Interestingly, IL-1β levels were shown to be significantly higher in tissue collected from patients undergoing elective TAA repair when compared to tissue collected from patients undergoing emergency repair for thoracic aortic dissection, suggesting that IL-1β plays a more important role in modulating TAA growth rather than aortic dissection [70].

Limited data on the involvement of other interleukins in TAA formation and aortic dissection have also been obtained, with levels of IL-11, IL-17 and IL-22, all of which display some pro-inflammatory activity, shown to be significantly elevated in aortic tissue and in the circulation of TAA patients presenting with an acute dissection [78,79]. Additionally, upregulation of the pro-inflammatory cytokine IL-3 signalling pathways and an increased in IL-3 positive cells in aneurysmal tissue has also been demonstrated in a mouse model of TAA dissection [80]. This limited evidence supports the role of IL-11, IL-22, IL-17 and IL-3 in aortic dissection, however, further investigations are needed to determine whether these interleukins have a role in the early stages of TAA development.”

  • The numbering of the sub-headings has been adjusted to take into account the two interleukin sections that have been combined. Additionally, there have been eight additional references added to the manuscript, hence the references have been added to the reference list and the numbering of the references has been adjusted to take into account the additional references.

  1. Page 12. At the end of conclusions section, the authors mentioned about novel therapeutic targets based on this review article, but this reviewer feels that it seems that the inflammatory responses in the genetically triggered TAA might be useful also for the development of novel biomarkers for early detection. If this is correct, the authors should mention not only treatment but also early detection.  

Response to reviewer: The concept of early detection has been incorporated into two locations within the manuscript:

  • the Abstract (Lines 24-25)

“The weight of evidence supports a role for inflammation in exacerbating the severity of TAA progression, expanding our understanding of the pathogenesis of TAA, identifying potential biomarkers for early detection of TAA, monitoring severity and progression and for defining potential novel therapeutic targets.”

  • and Conclusions (618-620).

“Overall, the relationship between inflammation and genetically triggered TAA constitutes fertile ground for expanding our understanding of the pathogenesis of TAA, identifying potential biomarkers for early detection of TAA, monitoring severity and progression and for defining potential novel therapeutic targets.”

Reviewer 2 Report

The authors wrote a comprehensive review on the etiology of thoracic aortic aneurysm and dissection and the contribution of myeloperoxidase-related oxidative stress. Although very well written, I have one major concern about the myeloperoxidase-related section. While the review is about thoracic aortic aneurysm, section 6.1 mostly contains information about links between MPO and other forms of aneurysmal disease that are genetically and pathomechanistically very different than TAA (and are not in depth described in the review since this is out of scope). Only two links between MPO and TAA remain if you exclude the former parts: MPO KO in MFS mice and increased MPO levels in a TAA cohort (of only 14 patients...). These very limited lines of evidence are subsequently linked to a pleiotropy of mechanisms involved in TAA formation, which seems utmost premature. 

Besides this major concern, I have the following additional remarks:

  • The title should be rephrased to "the role of inflammation and myeloperoxidase-related oxidative stress in the pathogenesis..." since not all aspects of oxidative stress are thoroughly covered. The focus is clearly on MPO-related oxidative stress.
  • Section 2.1: Collagen IV is not the most abundant one in the aorta => Specifically, in the aorta collagens type I and III constitute the largest portion. Whilst type III is more abundant in the medial layer, type I is the predominant form in the adventitia.
  • Section 3.1: Please also mention skeletal overgrowth and ectopia lentis, since both are very specific for MFS
  • Section 3.2: Please mention the characteristic LDS triad. Additionally, lens dislocation is an uncommon feature in LDS, while it is very common in MFS, so I would not mention it here. SMAD2 is also considered an LDS gene and should be added.
  • Section 3.5: More non-syndromic FTAAD genes exist, even not fitting within the VSMC pathway (e.g. LOX)
  • Section 3.6: Please expand this with some more information about the TAA GWAS (size, and top hits)
  • Section 4 introduction: reflect on the fact whether the inflammatory response occurs early in disease (driving factor) or late (consequence). Might the inflammatory response also be a "rescuing (unsuccesful)" compensatory event?
  • Line 223: extreme aortic dilatation but no aortic dissection
  • Section 4.5: To me it seems that this section should be incorporated in section 4.1. Seems the same but for a different model. 

Author Response

Reviewer 2

The authors wrote a comprehensive review on the etiology of thoracic aortic aneurysm and dissection and the contribution of myeloperoxidase-related oxidative stress. Although very well written, I have one major concern about the myeloperoxidase-related section. While the review is about thoracic aortic aneurysm, section 6.1 mostly contains information about links between MPO and other forms of aneurysmal disease that are genetically and pathomechanistically very different than TAA (and are not in depth described in the review since this is out of scope). Only two links between MPO and TAA remain if you exclude the former parts: MPO KO in MFS mice and increased MPO levels in a TAA cohort (of only 14 patients...). These very limited lines of evidence are subsequently linked to a pleiotropy of mechanisms involved in TAA formation, which seems utmost premature. 

Response to reviewer: We acknowledge that only relatively limited direct data are available for the involvement of MPO in genetically triggered TAA, which highlights the need for additional research in this area. We have altered the text to place the available data into an appropriate context in two locations within the manuscript:

  • We have clarified the text in the first paragraph of Section 6 (Lines 447-453)

“There have been several recent studies that have directly implicated MPO activity in the pathogenesis of aneurysm formation, and, specifically, limited animal and human data for direct involvement of MPO in TAA. Additionally, indirect evidence from a number of studies have demonstrated that MPO can exacerbate mechanisms that are known to be involved in TAA pathogenesis, such as damage to the ECM and DNA, activation of inflammatory signalling and an increase in endothelial dysfunction.”

  • We have clarified the Conclusions (Lines 615-616)

“All these pathological alterations lead to increased disruption of aortic wall homeostasis, resulting in a repeating cycle of more severe TAA development, which highlights the need for further investigations into the direct involvement of MPO in TAA.”

Besides this major concern, I have the following additional remarks:

  1. The title should be rephrased to "the role of inflammation and myeloperoxidase-related oxidative stress in the pathogenesis..." since not all aspects of oxidative stress are thoroughly covered. The focus is clearly on MPO-related oxidative stress.

Response to reviewer: We agree with the reviewer and have changed the title as suggested (Lines 2-4).

“The Role of Inflammation and Myeloperoxidase-related Oxidative Stress in the Pathogenesis of Genetically Triggered Thoracic Aortic Aneurysms”

  1. Section 2.1: Collagen IV is not the most abundant one in the aorta => Specifically, in the aorta collagens type I and III constitute the largest portion. Whilst type III is more abundant in the medial layer, type I is the predominant form in the adventitia.

Response to reviewer: Paragraph 2.1 outlines the extra-cellular matrix components of the aortic wall. We apologise for the confusion and have specifically clarified the major forms of collagen within each of the three layers. The specific major form of collagen within the intima is type IV, and we have clarified the paragraph by identifying that the major form of collagen within the media is type III and adventitia is type I (Lines 83-85).

“The tunica media consists of approximately 60 lamellae, consisting of elastin interspersed with VSMCs, embedded in extracellular-associated collagens (mainly type III) and glycosaminoglycans, which together form the contractile-elastic unit, whose function is to both bear and sense tension. The tunica adventitia is rich in collagen (mainly type I) and fibroblasts and contains the local blood (vasa vasorum) and neural supply.”

  1. Section 3.1: Please also mention skeletal overgrowth and ectopia lentis, since both are very specific for MFS

Response to reviewer: We have clarified the syndromal features of MFS in paragraph 3.1 (131-132).

Marfan syndrome (MFS) is an autosomal dominant inherited connective tissue disorder characterized by abnormalities in the eye (primarily ectopia lentis), disproportionate overgrowth of the skeleton and cardiovascular abnormalities.

  1. Section 3.2: Please mention the characteristic LDS triad.

Response to reviewer: We have clarified the characteristics of the LDS triad (Lines 146-147).

Diagnosis is usually based on the classic triad of arterial tortuosity, hypertelorism and wide or split uvula.

  1. Section 3.2: Additionally, lens dislocation is an uncommon feature in LDS, while it is very common in MFS, so I would not mention it here.

Response to reviewer: We have clarified the text by removed specific details of ocular manifestations (Lines 150-151).

Patients also experience skeletal, cutaneous and ocular abnormalities including scoliosis and pectus deformities, and velvety, thin translucent skin, blue sclera and retinal detachment.

  1. Section 3.2: SMAD2 is also considered an LDS gene and should be added.

Response to reviewer: We are aware that SMAD2 has been implicated as a cause of LDS, however, we note that the Aortopathy Working group of the NIH ClinGen framework lists SMAD2 as “insufficient evidence to support a definitive association, …. on the basis of only having 1 or 2 supporting publications [at 2018]”. Reference: J Am Coll Cardiol. 2018 August 07; 72(6): 605–615. doi:10.1016/j.jacc.2018.04.089 “Clinical Validity of Genes for Heritable Thoracic Aortic Aneurysm and Dissection”.

However, as suggested by the review we have incorporated SMAD2 mutations as a likely cause of LDS (Line 154).

The pathogenic variants responsible for LDS are genes that encode proteins within the TGF-b signalling pathway and include the TGF-b ligands (TGFB2 and TGFB3), their receptors (TGFBR1 and TGFBR2) and their intracellular signalling mediators (SMAD2/3) [29].”

  1. Section 3.5: More non-syndromic FTAAD genes exist, even not fitting within the VSMC pathway (e.g. LOX)

Response to reviewer: In section 3.5, we have clarified the diverse array of genes that are involved in fTAAD, including LOX (Lines 189-191).

Pathogenic variants in genes causing fTAAD may be involved in a diverse array of biological functions, including contractility of VSMCs (ACTA2, MYH11, MYLK, PRKG1), TGF-β signalling (TGFBR1, TGFBR2, TGFB2 and SMAD3) [41] and maintenance of ECM (LOX) [3].

  1. Section 3.6: Please expand this with some more information about the TAA GWAS (size, and top hits)

Response to reviewer: In section 3.6, we have clarified and provided more detail concerning the GWAS association with the FBN1 gene (Lines 203-205).

Notably, a genome wide association study comparing 765 subjects with sporadic TAA with 874 controls found associations with the 15q21.1 locus that includes the FBN1 gene, providing a potential link between the pathogenesis of MFS and sporadic TAA [43,44].

  1. Section 4 introduction: reflect on the fact whether the inflammatory response occurs early in disease (driving factor) or late (consequence). Might the inflammatory response also be a "rescuing (unsuccesful)" compensatory event?

Response to reviewer: In the Introduction to section 4 we have clarified the possible chronology of the inflammatory response in relation to the pathogenesis of TAA (Lines 211-215).

Notably, inflammation may occur early in the disease, thus being a driving factor, and/or later, as a consequence of the disease, where the response is a mal-adaptive, damaging compensatory process. A limitation of clinical studies of thoracic aortic tissue is the availability for harvest of only end-stage surgical tissue, rendering the chronology of the inflammatory process difficult to assess.

  1. Line 223: extreme aortic dilatation but no aortic dissection

Response to reviewer: In section 4.1 we have clarified that this mouse model of MFS demonstrated a correlation between MCP-1 expression and severe aortic dilatation (Lines 237-239).

Similarly, increased MCP-1 expression was found to correlate with more severe aortic dilatation in another MFS model mouse [51].

  1. Section 4.5: To me it seems that this section should be incorporated in section 4.1. Seems the same but for a different model. 

Response to reviewer: We have incorporated section 4.5 into section 4.1, and re-numbered the relevant sections and references (Lines 246-255).

Reviewer 3 Report

The manuscript is very well written, easy to read, and well structured, including pathogenesis, genetic aspects, and the role of inflammatory cells and mediators in TAA, to finally discuss the key issues of the review: the participation of inflammation, myeloperoxidase and oxidative stress in TAA pathogenesis. The review is closed with a figure that summarizes the evidence presented. The abstract provides precise information to get on the subject and the references are adequate, giving key information of the last ten years in the field, with a few older key references. Although there is information available about the role of oxidative stress in TAA, apparently there is no a recent revision on the subject, so I consider that this work may be useful for researchers in the field.

Author Response

Reviewer 3

The manuscript is very well written, easy to read, and well structured, including pathogenesis, genetic aspects, and the role of inflammatory cells and mediators in TAA, to finally discuss the key issues of the review: the participation of inflammation, myeloperoxidase and oxidative stress in TAA pathogenesis. The review is closed with a figure that summarizes the evidence presented. The abstract provides precise information to get on the subject and the references are adequate, giving key information of the last ten years in the field, with a few older key references. Although there is information available about the role of oxidative stress in TAA, apparently there is no a recent revision on the subject, so I consider that this work may be useful for researchers in the field.

Response to reviewer: We thank the reviewer for their supportive comments.

Round 2

Reviewer 2 Report

My remarks have adequately been addressed.